# How Deep is the Feature Analysis underlying Rapid Visual Categorization?

**Sven Eberhardt**[*]        **Jonah Cader**[*]        **Thomas Serre**
Department of Cognitive Linguistic & Psychological Sciences
Brown Institute for Brain Sciences
Brown University
Providence, RI 02818
{sven2,jonah_cader,thomas_serre}@brown.edu

## Abstract

Rapid categorization paradigms have a long history in experimental psychology: Characterized by short presentation times and speeded behavioral responses, these tasks highlight the efficiency with which our visual system processes natural object categories. Previous studies have shown that feed-forward hierarchical models of the visual cortex provide a good fit to human visual decisions. At the same time, recent work in computer vision has demonstrated significant gains in object recognition accuracy with increasingly deep hierarchical architectures. But it is unclear how well these models account for human visual decisions and what they may reveal about the underlying brain processes.

We have conducted a large-scale psychophysics study to assess the correlation between computational models and human behavioral responses on a rapid animal vs. non-animal categorization task. We considered visual representations of varying complexity by analyzing the output of different stages of processing in three state-of-the-art deep networks. We found that recognition accuracy increases with higher stages of visual processing (higher level stages indeed outperforming human participants on the same task) but that human decisions agree best with predictions from intermediate stages.

Overall, these results suggest that human participants may rely on visual features of intermediate complexity and that the complexity of visual representations afforded by modern deep network models may exceed the complexity of those used by human participants during rapid categorization.

## 1   Introduction

Our visual system is remarkably fast and accurate. The past decades of research in visual neuroscience have demonstrated that visual categorization is possible for complex natural scenes viewed in rapid presentations. Participants can reliably detect and later remember visual scenes embedded in continuous streams of images with exposure times as low as 100 ms [see 15, for review]. Observers can also reliably categorize animal vs. non-animal images (and other classes of objects) even when flashed for 20 ms or less [see 6, for review].

Unlike normal everyday vision which involves eye movements and shifts of attention, rapid visual categorization is assumed to involve a single feedforward sweep of visual information [see 19, for review] and engages our core object recognition system [reviewed in 5]. Interestingly, incorrect responses during rapid categorization tasks are not uniformly distributed across stimuli (as one would

---

[*]These authors contributed equally.

expect from random motor errors) but tend to follow a specific pattern reflecting an underlying visual strategy [1]. Various computational models have been proposed to describe the underlying feature analysis [see 2, for review]. In particular, a feedforward hierarchical model constrained by the anatomy and the physiology of the visual cortex was shown to agree well with human behavioral responses [16].

In recent years, however, the field of computer vision has championed the development of increasingly deep and accurate models – pushing the state of the art on a range of categorization problems from speech and music to text, genome and image categorization [see 12, for a recent review]. From AlexNet [11] to VGG [17] and Microsoft CNTK [8], over the years, the ImageNet Large Scale Visual Recognition Challenge (ILSVRC) has been won by progressively deeper architectures. Some of the ILSVRC best performing architectures now include 150 layers of processing [8] and even 1,000 layers for other recognition challenges [9] – arguably orders of magnitude more than the visual system (estimated to be $\mathcal{O}(10)$, see [16]). Despite the absence of neuroscience constraints on modern deep learning networks, recent work has shown that these architectures explain neural data better than earlier models [reviewed in 20] and are starting to match human level of accuracy for difficult object categorization tasks [8].

It thus raises the question as to whether recent deeper network architectures better account for speeded behavioral responses during rapid categorization tasks or whether they have actually become too deep – instead deviating from human responses. Here, we describe a rapid animal vs. non-animal visual categorization experiment that probes this question. We considered visual representations of varying complexity by analyzing the output of different stages of processing in state-of-the-art deep networks [11, 17]. We show that while recognition accuracy increases with higher stages of visual processing (higher level stages indeed outperforming human participants for the same task) human decisions agreed best with predictions from intermediate stages.

## 2   Methods

**Image dataset**   A large set of (target) animal and (distractor) non-animal stimuli was created by sampling images from ImageNet [4]. We balanced the number of images across basic categories from 14 high-level synsets, to curb biases that are inherent in Internet images. (We used the invertebrate, bird, amphibian, fish, reptile, mammal, domestic cat, dog, structure, instrumentation, consumer goods, plant, geological formation, and natural object subtrees.) To reduce the prominence of low-level visual cues, images containing animals and objects on a white background were discarded. All pictures were converted to grayscale and normalized for illumination. Images less than 256 pixels in either dimension were similarly removed and all other images were cropped to a square and scaled to $256 \times 256$ pixels. All images were manually inspected and mislabeled images and images containing humans were removed from the set ($\sim 17\%$ of all images). Finally, we drew stimuli uniformly (without replacement) from all basic categories to create balanced sets of 300 images. Each set contained 150 target images (half mammal and half non-mammal animal images) and 150 distractors (half artificial objects and half natural scenes). We created 7 such sets for a total of 2,100 images used for the psychophysics experiment described below.

**Participants**   Rapid visual categorization data was gathered from 281 participants using the Amazon Mechanical Turk (AMT) platform (`www.mturk.com`). AMT is a powerful tool that allows the recruitment of massive trials of anonymous workers screened with a variety of criteria [3].

All participants provided informed consent electronically and were compensated $4.00 for their time ($\sim$ 20–30 min per image set, 300 trials). The protocol was approved by the University IRB and was carried out in accordance with the provisions of the World Medical Association Declaration of Helsinki.

**Experimental procedure**   On each trial, the experiment ran as follows: On a white background (1) a fixation cross appeared for a variable time (1,100–1,600 ms); (2) a stimulus was presented for 50 ms. The order of image presentations was randomized. Participants were instructed to answer as fast and as accurately as possible by pressing either the "S" or "L" key depending on whether they saw an animal (target) or non-animal (distractor) image. Key assignment was randomized for each participant.

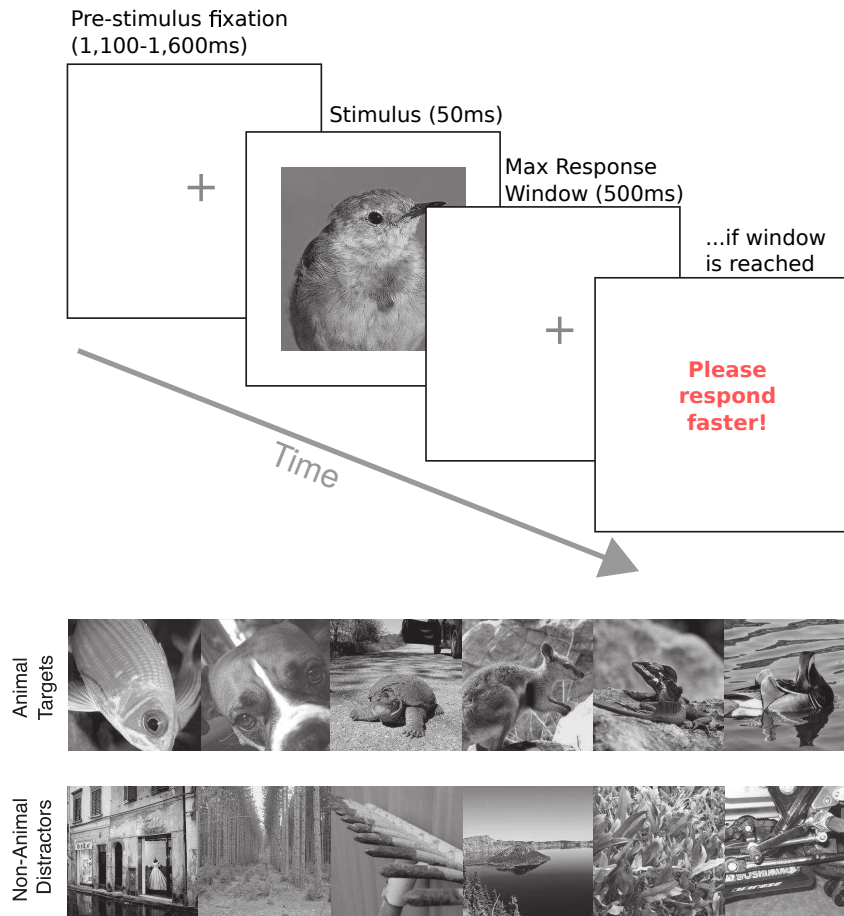

Figure 1: **Experimental paradigm and stimulus set**: (top) Each trial began with a fixation cross (1,100–1,600 ms), followed by an image presentation (∼ 50 ms). Participants were forced to answer within 500 ms. A message appeared when participants failed to respond in the allotted time. (bottom) Sample stimuli from the balanced set of animal and non-animal images (n=2,100). A fast answer time paradigm was used in favor of masking to avoid possible performance biases between different classes caused by the mask [6, 15].

Participants were forced to respond within 500 ms (a message was displayed in the absence of a response past the response deadline). In past studies, this has been shown to yield reliable behavioral data [e.g. 18]. We have also run a control to verify that the maximum response time did not affect qualitatively our results.

An illustration of the experimental paradigm is shown in Figure 1. At the end of each block, participants received feedback about their accuracy. An experiment started with a short practice during which participants were familiarized with the task (stimulus presentation was slowed down and participants were provided feedback on their response). No other feedback was provided to participants during the experiment.

We used the psiTurk framework [13] combined with custom javascript functions. Each trial (i.e., fixation cross followed by the stimulus) was converted to a HTML5-compatible video format to provide the fastest reliable presentation time possible in a web browser. Videos were generated to include the initial fixation cross and the post-presentation answer screen with the proper timing as described above. Videos were preloaded before each trial to ensure reliable image presentation times over the Internet.

We used a photo-diode to assess the reliability of the timing on different machines including different OS, browsers and screens and found the timing to be accurate to ∼ 10 ms. Images were shown at a

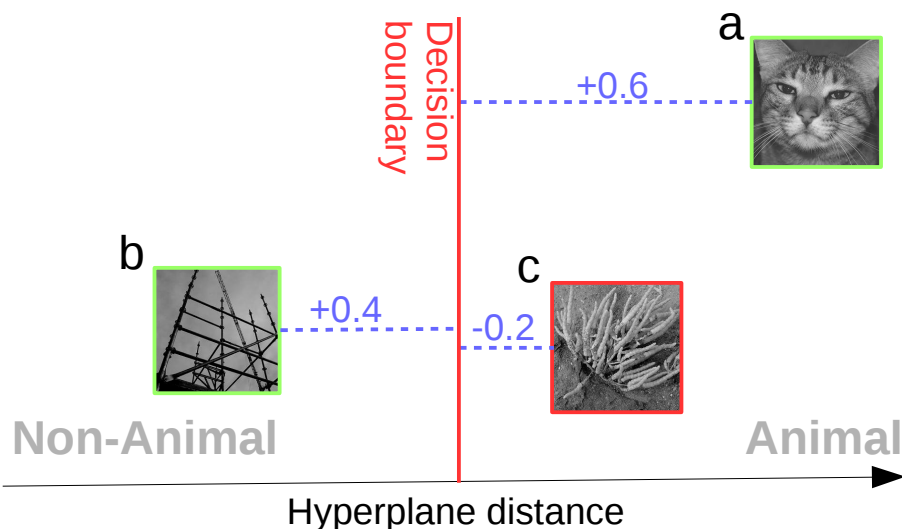

Figure 2: **Model decision scores**: A classifier (linear SVM) is trained on visual features corresponding to individual layers from representative deep networks. The classifier learns a decision boundary (shown in red) that best discriminates target/animal and distractor/non-animal images. Here, we consider the signed distance from this decision boundary (blue dotted lines) as a measure of the model's confidence on the classification of individual images. A larger distance indicates higher confidence. For example, while images (a) and (b) are both correctly classified, the model's confidence for image (a) correctly classified as animal is higher than that of (b) correctly classified as non-animal. Incorrectly classified images, such as (c) are assigned negative scores corresponding to how far onto the wrong side of the boundary they fall.

resolution of $256 \times 256$. We estimate this to correspond to a stimulus size between approximately $5^o - 11^o$ visual angle depending on the participants' screen size and specific seating arrangement.

The subjects pool was limited to users connections from the United States using either the Firefox or Chrome browser on a non-mobile device. Subjects also needed to have a minimal average approval rating of 95% on past Mechanical Turk tasks.

As stated above, we ran 7 experiments altogether for a total of 2,100 unique images. Each experiment lasted 20-30 min and contained a total of 300 trials divided into 6 blocks (50 image presentations / trials each). Six of the experiments followed the standard experimental paradigm described above (1,800 images and 204 participants). The other 300 images and 77 participants were reserved for a control experiment in which the maximum reaction time per block was set to 500 ms, 1,000 ms, and 1,500 ms for two block each. (See below.)

**Computational models** We tested the accuracy of individual layers from state-of-the-art deep networks including AlexNet [11], VGG16 and VGG19 [17]. Feature responses were extracted from different processing stages (Caffe implementation [10] using pre-trained weights). For fully connected layers, features were taken as is; for convolutional layers, a subset of 4,096 features was extracted via random sampling. Model decisions were based on the output of a linear SVM (scikit-learn [14] implementation) trained on 80,000 ImageNet images ($C$ regularization parameter optimized by cross-validation). Qualitatively similar results were obtained with regularized logistic regression. Feature layer accuracy was computed from SVM performance.

Model confidence for individual test stimuli was defined as the estimated distance from the decision boundary (see Figure 2). A similar confidence score was computed for human participants by considering the fraction of correct responses for individual images. Spearman's rho rank-order correlations ($r_s$) was computed between classifier confidence outputs and human decision scores. Bootstrapped 95% confidence intervals (CIs) were calculated on human-model correlation and human classification scores. Bootstrap runs (n=300) were based on 180 participants sampled with

replacement from the subject pool. CIs were computed by considering the bottom 2.5% and top 97.5% values as upper and lower bounds.

## 3  Results

We computed the accuracy of individual layers from commonly used deep networks: AlexNet [11] as well as VGG16 and VGG19 [17]. The accuracy of individual layers for networks pre-trained on the ILSVRC 2012 challenge (1,000 categories) is shown in Figure 3 (a). The depth of individual layers was normalized with respect to the maximum depth as layer depth varies across models. In addition, we selected VGG16 as the most popular state-of-the-art model and fine-tuned it on the animal vs. non-animal categorization task at hand. Accuracy for all models increased monotonically (near linearly) as a function of depth to reach near perfect accuracy for the top layers for the best networks (fine-tuned VGG16). Indeed, all models exceeded human accuracy on this rapid animal vs. non-animal categorization task. Fine-tuning did improve test accuracy slightly from 95.0% correct to 97.0% correct on VGG16 highest layer, but the performance of all networks remained high in the absence of any fine-tuning.

To benchmark these models, we assessed human participants' accuracy and reaction times (RTs) on this animal vs. non-animal categorization task. On average, participants responded correctly with an accuracy of 77.4% ($\pm$ 1.4%). These corresponded to an average d' of 1.06 ($\pm$ 0.06). Trials for which participants failed to answer before the deadline were excluded from the evaluation (13.7% of the total number of trials). The mean RT for correct responses was 429 ms ($\pm$ 103 ms standard deviation). We computed the minimum reaction time *MinRT* defined as the first time bin for which correct responses start to significantly outnumber incorrect responses [6]. The MinRT is often considered a floor limit for the entire visuo-motor sequence (feature analysis, decision making, and motor response) and could be completed within a temporal window as short as 370 ms $\pm$ 75 ms. We computed this using a binomial test (p < 0.05) on classification accuracy from per-subject RT data sorted into 20 ms bins and found the median value of the corresponding distribution.

Confidence scores for each of the 1,800 (animal and non-animal) main experiment images were calculated for human participants and all the computational models. The resulting correlation coefficients are shown in Figure 3 (b). Human inter-subject agreement, measured as Spearman's rho correlation between 1,000 randomly selected pairs of bootstrap runs, is at $\rho = 0.74$ ($\pm$ 0.05%). Unlike individual model layer accuracy which increases monotonically, the correlation between these same model layers and human participants picked for intermediate layers and decreased for deeper layers. This drop-off is stable across all tested architectures and started around at 70% of the relative model depth. For comparison, we re-plotted the accuracy of the individual layers and correlation to human participants for the fine-tuned VGG16 model in Figure 3 (c). The drop-off in correlation to human responses begins after layer conv5_2, where the correlation peaks at $0.383 \pm 0.026$. Without adjustment, i.e. correlating the answers including correctness, the peak lies at the same layer at $0.829 \pm 0.008$ (see supplement B for graph).

Example images in which humans and VGG16 top layer disagree are shown in Figure 4. The model typically outperforms humans on elongated animals such as snakes and worms, as well as camouflaged animals and when objects are presented in an atypical context. Human participants outperform the model on typical, iconic illustrations such as a cat looking directly at the camera.

We verified that the maximum response time (500 ms) allowed did not qualitatively affect our results. We ran a control experiment (77 participants) on a set of 300 images where we systematically varied the maximum response time available (500 ms, 1,000 ms and 2,000 ms). We evaluated differences in categorization accuracy using a one-way ANOVA with Tukey's HSD for post-hoc correction. The accuracy increased significantly from 500 to 1,000 ms (from 74 % to 84 %; p < 0.01). However, no significant difference was found between 1,000 and 2,000 ms (both $\pm$ 84%; p > 0.05). Overall, we found no qualitative difference in the observed pattern of correlation between human and model decision scores for longer response times (results in supplement A). We found an overall slight upward trend for both intermediate and higher layers for longer response times.

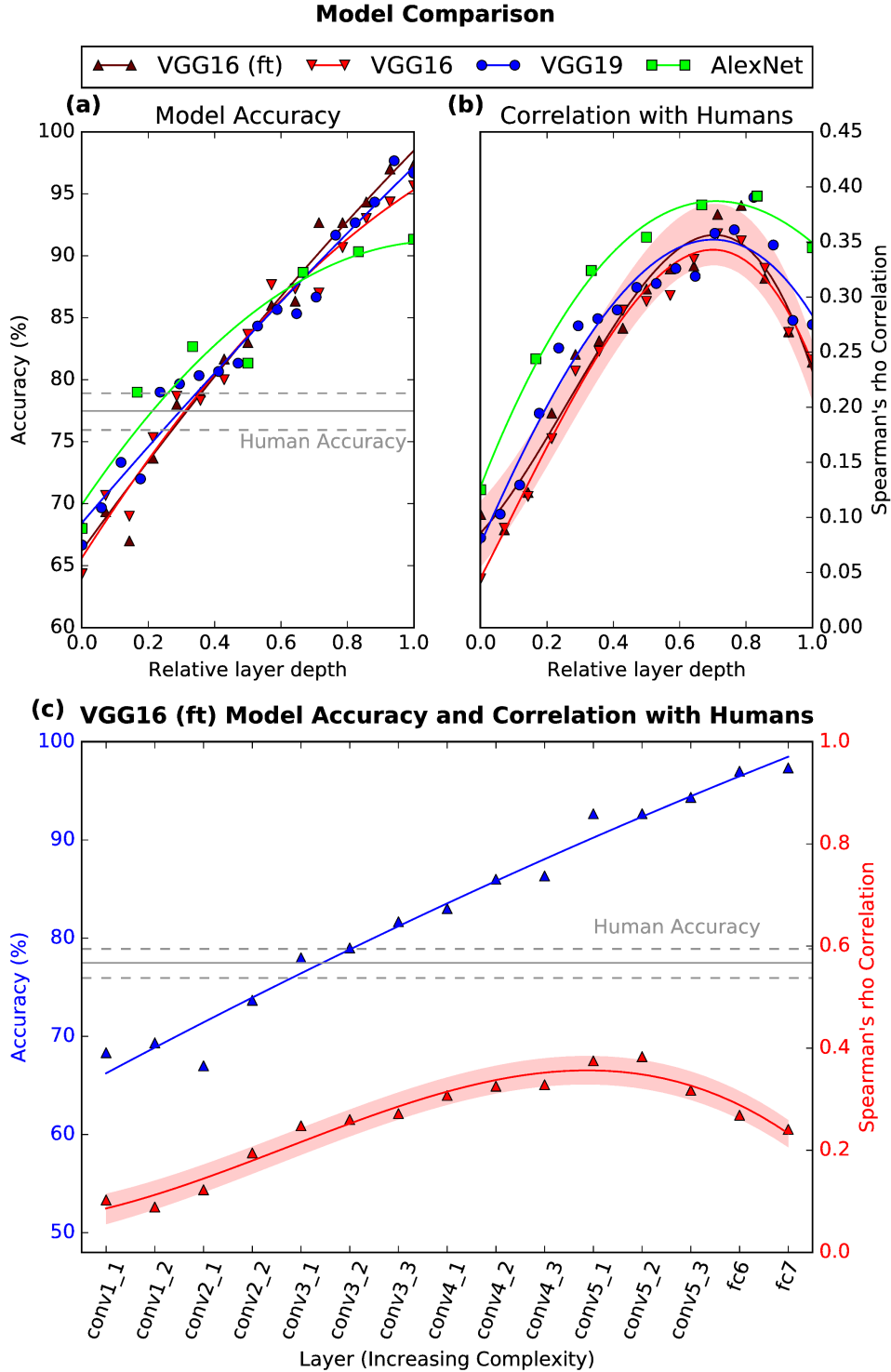

Figure 3: **Comparison between models and human behavioral data:** (a) Accuracy and (b) correlation between decision scores derived from various networks and human behavioral data plotted as a function of normalized layers depth (normalized by the maximal depth of the corresponding deep net). (c) Superimposed accuracy and correlation between decision scores derived from the best performing network (VGG16 fine-tuned (*ft*) for animal categorization) and human behavioral data plotted as a function of the raw layers depth. Lines are fitted polynomials of 2nd (accuracy) and 3rd (correlation) degree order. Shaded red background corresponds to 95% CI estimated via bootstrapping shown for fine-tuned VGG16 model only for readability. Gray curve corresponds to human accuracy (CIs shown with dashed lines).

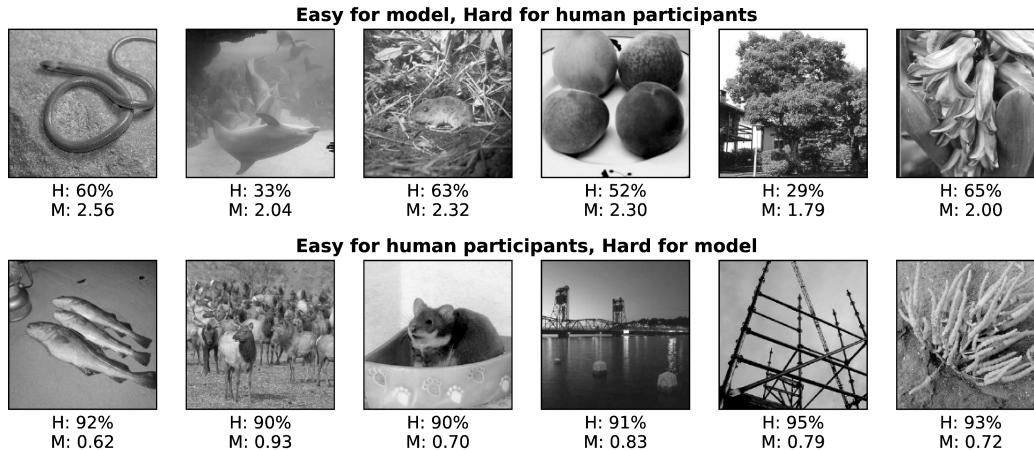

Figure 4: **Sample images where human participants and model (VGG16 layer fc7) disagree.** H: Average human decision score (% correct). M: Model decision score (distance to decision boundary).

# 4   Discussion

The goal of this study was to perform a computational-level analysis aimed at characterizing the visual representations underlying rapid visual categorization.

To this end, we have conducted a large-scale psychophysics study using a fast-paced animal vs. non-animal categorization task. This task is ecologically significant and has been extensively used in previous psychophysics studies [reviewed in 6]. We have considered 3 state-of-the-art deep networks: AlexNet [11] as well as VGG16 and VGG19 [17]. We have performed a systematic analysis of the accuracy of these models' individual layers for the same animal/non-animal categorization task. We have also assessed the agreement between model and human decision scores for individual images.

Overall, we have found that the accuracy of individual layers consistently increased as a function of depth for all models tested. This result confirms the current trend in computer vision that better performance on recognition challenges is typically achieved by deeper networks. This result is also consistent with an analysis by Yu et al. [21], who have shown that both sparsity of the representation and distinctiveness of the matched features increase monotonously with the depth of the network.

However, the correlation between model and human decision scores peaked at intermediate layers and decreased for deeper layers. These results suggest that human participants may rely on visual features of intermediate complexity and that the complexity of visual representations afforded by modern deep network models may exceed those used by human participants during rapid categorization. In particular, the top layers (final convolutional and fully connected), while showing an improvement in accuracy, no longer maximize correlation with human data. Whether this result is based on the complexity of the representation or invariance properties of intermediate layers remains to be investigated. It should be noted that a depth of $\sim 10$ layers of processing has been suggested as an estimate for the number of processing stages in the ventral stream of the visual cortex [16].

How then does the visual cortex achieve greater depth of processing when more time is allowed for categorization? One possibility is that speeded categorization reflects partial visual processing up to intermediate levels while longer response times would allow for deeper processing in higher stages. We compared the agreement between model and human decisions scores for longer response times (500 ms, 1,000 ms and 2,000 ms). While the overall correlation increased slightly for longer response times, this higher correlation did not appear to differentially affect high- vs. mid-level layers.

An alternative hypothesis is that greater depth of processing for longer response times is achieved via recurrent circuits and that greater processing depth is achieved through time. The fastest behavioral responses would thus correspond to bottom-up / feed-forward processing. This would be followed by re-entrant and other top-down signals [7] when more time is available for visual processing.

## Acknowledgments

We would like to thank Matt Ricci for his early contribution to this work and further discussions. This work was supported by NSF early career award [grant number IIS-1252951] and DARPA young faculty award [grant number YFA N66001-14-1-4037]. Additional support was provided by the Center for Computation and Visualization (CCV).

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
