[Supplementary Material · Supplement_Eberhardt_et_al_NIPS_2016.pdf]

# 1 Supplementary Figures

Supplement A: **Comparison between models and human behavioral data for different maximum answer times:** Accuracy and correlation between decision scores for different layers in the VGG16 model, plotted for short (500ms) and longer (1000ms and 1500ms) maximum answer time trials. Lines are fitted polynomials of 1st (accuracy) and 3rd (correlation) degree order. Gray curve corresponds to human accuracy over all trials (CIs shown with dashed lines). While correlation is slightly higher for longer presentation times, the correlation peak layer is independent of the timing.

**Supplement B: Comparison between models and human behavioral data by correlating the correctness score:** (a) Accuracy and (b) correlation between correctness scores derived from various networks and human behavioral data plotted as a function of normalized layers depth (normalized by the maximal depth of the corresponding deep net). (c) Superimposed accuracy and correctness score correlation between decision scores derived from the best performing network (VGG16 fine-tuned (*ft*) for animal categorization) and human behavioral data plotted as a function of the raw layers depth. Lines are fitted polynomials of 2nd (accuracy) and 3rd (correlation) degree order. Gray curve corresponds to human accuracy (CIs shown with dashed lines). Compared to figure 3 (correlation of decision scores), correlations are generally much higher because they include the correctness of the label. The correlation falloff at higher layers is also less pronounced; we assume that the larger number of correct labels compensates for the reduction in correlation at higher layers.