[Reviews · NeurIPS 2016]

Reviewer 1

Summary

The paper compares performance of deep networks to psychophysical performance of humans performing classification on the same images. From the correlation in performance it is claimed that human performance is best modeled based on intermediate levels of a deep net.

Qualitative Assessment

Overall this is an interesting study, and I do appreciate the method of using Mechanical Turk to collect data. As far as I know the result is novel, that humans correlate most strongly with intermediate layers of a variety of different deep nets. I think the Discussion section is a bit to grandiose in claiming that layer 10 of their fine-tuned VGG-16 model presents some sort of supporting evidence of the Serre et al claim about how many processing stages humans have. I also would have liked to see an investigation of what the representations at 70% total depth were for the different networks. It would have supported their argument better if they could have asserted that the features at 70% depth in a smaller network, such as AlexNet corresponded strongly with the features found in a larger net, such as VGG-19. Specific comments: As far as I could tell they did not use a mask for the human subjects. Why? If we assume that the mask serves the purpose it is intended to serve, then it should stop contributions of recurrent and top-down processing from influencing the human's response. I think using a mask would increase support for their argument that error correlations can be used to assess the "depth" of the human recognition system. The authors compared the human results to results from three staple deep nets: AlexNet, VGG16, and VGG19. For convolutional layers they grab a random sample of features from each layer. They fix the number of features they use, which I find a little concerning. Given that deeper layers in these networks have fewer features, if they fix the sample size then they are going to get a more complete set of features as they ascend the hierarchy, and thus a more complete description of the layer's representation. Given the features from a certain Deep Net layer, they use a SVM to perform the 2AFC task. This surprises me, I would think that they would have just used a couple of fully connected layers as is done in the ImageNet challenge. They find that the humans and deep nets have the highest response correlation when using features from layers at 70% of the model's depth. It seems like this is consistent across models, which is a bit confusing. The conclusion they draw from the result is that humans use intermediate-level features. However, it is based on the assumption that the features at whatever layer corresponds to 70% of the model's depth are uniform across models. They do not discuss this directly, and I would be very interested to see if it is true at all. Does modifying the total model depth have any influence on the features learned at a relative depth? They further note that the maximum correlation observed with one of their models (fined tuned VGG16) occurs with the 10th layer of processing, which is about how many "processing stages" Serre, Oliva, & Poggio claim are in the ventral stream. I find this whole argument misleading, given that the specific layer depth that correlates with humans is going to vary monotonically with the total depth of the network. If you consider their result that the highest correlation is always at around 70% of the total depth, it seems reasonable to just choose a network that has the correct number of total layers to match the Serre et al.'s findings. Also, they claim that they use VGG-16 because it is the best performing network, but that is not true according to Figure 3a. VGG-19 performs best out of the box, only after fine-tuning VGG-16 do they do the best. They do not comment on how well a fine-tuned VGG-19 net would do.

Confidence in this Review

2-Confident (read it all; understood it all reasonably well)


Reviewer 2

Summary

The paper conducts a psychophysics study to assess the correlation between human performance on rapid visual categorization tasks and that predicted by features learnt by recent deep neural networks when trained on large data. In particular, the study evaluated correlation vis-a-vis features learnt at various depths by such networks. The study concludes that with a suggestion that the feature representations used by the human visual system may be of intermediate complexity, similar to those learnt by deep learning networks at intermediate depths (around layer 10).

Qualitative Assessment

The paper addresses an important question - regarding the complexity of the features computed by the human visual system for rapid visual categorization. The paper is also written well and as far as I can tell, the experiment conducted is a novel one and may add value to the knowledge regarding the human visual system. However, I am not convinced about the value of the particular experiments conducted and the results reported in the paper. In particular, (a) There is a mismatch between the relative depths at which these networks obtain human level accuracy and the relative depths at which the correlation with human performance on individual images peaks. This is an important artifact that goes unexplained in the paper. (b) Further, the correlation with human performance peaks at 35 percent and hence I am not sure if the features computed at those depths or the complexity of these features says anything about (or predicts) the kinds of features computed by the human visual system. (c) It is well known that the human neural mechanisms as well as the network topology are very different from those used by ANNs. (d) Further, there is an issue with response time cutoff being kept at 500 ms. As discussed in lines 192-196, increasing the response time to 1000 ms increases human accuracy significantly. Though this does not differentially impact conclusions about the relative depth at which the correlation peaks, it would perhaps reduce the artifact mentioned in point (a) here. These performance curves are not presented. In light of all of this, I am not sure there is a useful takeaway from the experiments and the results reported in the paper. I am willing to be persuaded otherwise by the authors in the rebuttal phase.

Confidence in this Review

2-Confident (read it all; understood it all reasonably well)


Reviewer 3

Summary

This paper compares the performance of humans and deep networks in fast categorization. The aim is not to improve performance but rather, provide some insight as to the extent to which natural processes in the brain may match current vision recognition architectures. The meat of the paper is an experiment of animal vs. non-animal fast categorization. On the human side, the experiment is run on MTurk. On the machine side, several deep networks (AlexNet, VGG16 and VGG19) are trained and the representations learned at each layer are extracted and used as input for classification with a linear classifier (SVM or logistic regression). Correlating human and machine decision scores show that human decision profile corresponds best to intermediate representations of deep networks, and that the extra layers afford artificial nets an added complexity (that indeed may explain their super-human performance).

Qualitative Assessment

This is an interesting paper for people who are interested in the link between artificial nets and human brain functioning. The paper reads well, is clear. I would expect a subset of the nips audience to be interested in those results. The discussion sees the fact that the maximum correlation is at about 10 layers, as consistent with the number of processing stages found in the ventral stream of the visual cortex in ref. 16 (Serre et al 2007). I find this claim a bit strange -- "around 10" can be 70% of 19, which is 13.3 and a margin of about one third, which doesn't seem much of a claim for networks of a maximum depth of 19 layers -- it all seems to be the same order of magnitude. What would the result be with even deeper networks, like the approx. 150 layers of this year's ImageNet? Would the authors expect the maximum correlation to then be at around 10% of the total layer depth?

Confidence in this Review

3-Expert (read the paper in detail, know the area, quite certain of my opinion)


Reviewer 4

Summary

The work describes a classic psych-phsyical experiment, where the subject is asked to answer the question if an animal can be seen in the picture. Results are then compared with the output of several out of the box available deep learning systems. The decision process is then done by using a linear support vector machine. Main result of the paper is a comparison of human results with the performance of the ANNs for each layer, where human performance is outperformed at 30% of the layer depth of the used depth of the deep learning algorithms.

Qualitative Assessment

One issue is that the paper still lacks real clarity one the meaning of the relative layer depth, that is how that corresponds to what kind computational processing complexity. So still it is hard to interpret the data, i.e. that humans are outperformed on what level of the layered natwork doing what.

Confidence in this Review

1-Less confident (might not have understood significant parts)


Reviewer 5

Summary

The paper provided a large-scale experiment to measure visual classification performance of human being, to compare with state-of-the art deep learning methods. The methodology to assess the visual processing mechanism is well organized. However, there is no novel techniques proposed in this paper. Psychologically, it is found that human's visual processing may be comparable to 8-12 layers of current deep networks in terms of correlation.

Qualitative Assessment

There is no discussion on the reason why the artificial networks with layers having peak correlation can outperform human's behavior. There is no counterintuitive result. In other words, it is better to add more important insights to assist further understanding of psychological and technological advancement. It is also better to propose a bit new signal processing method rather than using existing methods.

Confidence in this Review

2-Confident (read it all; understood it all reasonably well)